# Single Surgeon versus Co-Surgeons in Primary Total Joint Arthroplasty: Does “Two Is Better than One” Apply to Surgeons?

**DOI:** 10.3390/jpm12101683

**Published:** 2022-10-09

**Authors:** Tal Frenkel Rutenberg, Maria Vitenberg, Efrat Daglan, Assaf Kadar, Shai Shemesh

**Affiliations:** 1Sackler Faculty of Medicine, Tel Aviv University, Tel Aviv 6997801, Israel; 2Orthopedic Department, Rabin Medical Center, Beilinson Hospital, Petah-Tikva 4941492, Israel

**Keywords:** arthroplasty, hip, knee, attending, resident, assisting surgeon

## Abstract

Background: As the demand for total joint arthroplasties (TJA) increases steadily, so does the pressure to train future surgeons and, at the same time, achieve optimal outcomes. We aimed to identify differences in operative times and short-term surgical outcomes of TJAs performed by co-surgeons versus a single attending surgeon. Methods: A retrospective analysis of 597 TJAs, including 239 total hip arthroplasties (THAs) and 358 total knee arthroplasties (TKAs) was conducted. All operations were performed by one of four fellowship-trained attending surgeons as the primary surgeon. The assisting surgeons were either attendings or residents. Results: In 51% of THA and in 38% of TKA, two attending surgeons were scrubbed in. An additional scrubbed-in attending was not found to be beneficial in terms of surgical time reduction or need for revision surgeries within the postoperative year. This was also true for THAs and for TKAs separately. An attending co-surgeon was associated with a longer hospital stay (*p* = 0.028). Surgeries performed by fewer surgeons were associated with a shorter surgical time (*p* = 0.036) and an increased need for blood transfusion (*p* = 0.033). Neither the rate of intraoperative complications nor revisions differed between groups, regardless of the number of attending surgeons scrubbed in or the total number of surgeons. Conclusion: A surgical team comprised of more than a single attending surgeon in TJAs was not found to reduce surgical time, while the participation of residents was not related with worse patient outcomes.

## 1. Introduction

Total knee arthroplasty (TKA) and total hip arthroplasty (THA) are both successful surgical interventions, proved to improve quality of life, with the number of procedures substantially growing each year [1]. As the demand for these surgeries steadily increases, so does the need to train surgeons. However, as the need to increase the number of operations performed motivates hospitals to present favorable data regarding reduced surgical duration and complication rates as well as increased patient satisfaction, the education of young surgeons can become compromised [2,3,4].

Several risk factors were found to be associated with prolonged operative time in total joint arthroplasty (TJA) surgeries, among which are: male gender, white race, increased body mass index (BMI), smoking, the number of co-morbidities, the use of general anesthesia and the experience level of the surgeon [5,6,7,8]. Prolonged operative time in TJAs was found to be a risk factor for anemia necessitating blood transfusions, proximal deep vein thrombosis (DVT), renal insufficiency, sepsis, urinary tract infections, wound dehiscence and surgical site infections (SSI) [9,10,11].

Several studies have examined the influence of orthopedic surgical trainees on the outcome of TJA surgeries. Most studies found that increased trainee involvement increases operative time, yet this was not found to be related with higher rates of intraoperative or postoperative complications, or with worse patient-reported outcomes or satisfaction rates by most [2,3,4,12,13]. To our knowledge, only a single study compared the effect of two attending surgeons versus a single attending surgeon aided by a resident or physician assistant on the operative time and surgical outcomes and found no improvement in either. However, the study cohort was very small and included only 40 patients in the two attendings group [3].

This study aims to identify differences in operative times and short-term surgical outcomes of TJA procedures performed by co-surgeons versus a single attending surgeon at a single institution. We hypothesized that TJA procedures performed by an attending surgeon assisted by an attending co-surgeon would shorten the surgical time and lower the complication and early revision rates.

## 2. Materials and Methods

### 2.1. Study Design

We performed a single-center retrospective analysis of consecutive patients undergoing primary TJA (THA and TKA) between 2013–2017, after obtaining ethical approval from the institutional review board.

Operating room (OR) case logs were used to identify our cohort of patients. Primary THAs and TKAs were identified by ICD-9-CM (International Classification of Diseases, Ninth Revision, Clinical Modification) code 81.51 and 81.54, respectively. We subsequently excluded bilateral TJAs, conversion to TJA, revision surgeries and surgeries with more than two attending surgeons involved. All operations were performed by one of four fellowship-trained attending surgeons as the primary surgeon. The assisting surgeon was either another attending surgeon (from the four mentioned above) or a resident (who only assisted). In some surgeries done by two attending surgeons, a resident or two were also scrubbed in, as a third and fourth member of the team. In a few surgeries, additional residents were also scrubbed in. The allocation of surgeons to the OR was based upon staff availability at the time, and not upon the surgical case at hand. All surgeries were performed in the public medical system. Physiotherapy was performed daily on weekdays, and patients were discharged when the attending surgeon found them fit, after several courses of physiotherapy.

Patients were divided into groups according to the surgical procedure performed, either TKA or THA. For each surgical group, two analyses were performed, one comparing outcomes of surgeries in which only one attending surgeon was scrubbed in, compared with surgeries in which two attending surgeons took part, and a second comparing the total number of surgeons in the procedure (two, three or four). Of note, the surgical team included only medical doctors, and not physician assistants, as accustomed in our healthcare system. A single scrubbed nurse and a single circulating nurse were present as well in all cases.

### 2.2. Variables and Measurements

The operative case logs included the following data elements: surgeon(s) seniority, exact composition of the surgical team, surgery incision start and end time (length of surgery), type of anesthesia, patient’s American Society of Anesthesiologists (ASA) score, primary procedure type and intraoperative complications. From the final dataset of patients, retrospective chart review was conducted to extract patient age, gender, body mass index (BMI), medical history (which included current smoking, diabetes mellitus (DM), rheumatoid arthritis (RA) and malignancy (current or past), length of stay (LOS), need for blood transfusions and the number of packed cells administered, and revisions for any cause within the first postoperative year.

### 2.3. Procedures

Prior to TKA, all patients completed coronal and sagittal weight-bearing radiographs as well as a long-leg coronal radiograph, for purposes of preoperative planning. Three knee systems were used: Unity kneeTM (Corin, Cirencester, UK), Attune^®^ (DePuy Synthes, Warsaw, Indiana) and Sigma^®^ (DePuy Synthes, Warsaw, Indiana). In all cases, mechanically aligned TKA was performed through a standard medial parapatellar approach, and cement was used. The surgical implant and design (cruciate retaining or posterior stabilized) was chosen based on the attending surgeon’s preference, as was the use of tourniquet. Similarly, patella resurfacing was performed according to the attending surgeon’s preference. The standard extramedullary guide was used for the proximal tibia cut, whereas intramedullary guides were used to facilitate the femoral cuts.

Prior to THA, all patients completed pelvic x-rays with a reference ball. Surgical approach (posterior approach (43.1%), lateral approach (16.7%), direct anterior (37.2%), anterolateral (1.3%) or posterolateral (1.7%) approach) and implants were chosen according to the attending surgeon’s preference. Cement was used in 79 THA procedures, according to the surgeon’s preference.

### 2.4. Outcomes

The primary outcome was the operative time. Secondary outcome measures were intraoperative complications, LOS, blood transfusions during the hospital stay and 1-year revision rates (in our hospital and others in the country, searched through a shared computerized medical record).

### 2.5. Statistical Analysis

Descriptive statistics were based on percentages and frequencies for categorical variables and for continuous variables. Continuous variables are presented as mean and standard deviation (SD). Quantitative variables are presented as absolute and relative frequencies. Fisher’s exact tests was used for comparison of categorical variables, and Student’s t-test for continuous variables. For variables with more than two subcategories, ANOVA (analysis of variance) was conducted to detect differences between groups. All reported *p*-values are two-tailed. Statistical significance is defined as *p* < 0.05. All analyses were performed using SPSS (SPSS 25.0, IBM Inc., Somers, NY, USA).

## 3. Results

In total, 705 TJAs were performed during the study period. Of these, 108 TJAs were excluded (31 revisions, 70 surgeries which were performed due to fractures and seven surgeries which were performed by more than two attending surgeons). Thus, 597 primary TJAs were included in the analysis: 239 THAs and 358 TKAs. In 51% of THAs and in 38% of TKAs, two attending surgeons were scrubbed in. The four senior surgeons were equally distributed between the two groups (*p* = 0.776 for the entire cohort, *p* = 0.42 for TKA and *p* = 0.57 for THA). In the TKA cohort, the three different knee systems used were equally distributed between the single attending and two attendings groups (*p* = 0.178).

An additional scrubbed-in attending surgeon was not found to be beneficial for TJA in terms of reduced surgical duration (1:41 ± 00:29 min vs. 1:37 ± 00:28 min (*p* = 0.152)), complications within surgery (nine (2.7%) vs. six (2.3%) (*p* = 1)), blood transfusion required (0.12 ± 0.5 vs. 0.06 ± 0.34 units per patient (*p* = 0.119)) or revision surgeries for any cause within the postoperative year (four (1.2%) vs. three (1.2%) (*p* = 1)), when comparing two attending surgeons with a single attending surgeon scrubbed in (Table 1). This was also true for THAs and for TKAs separately (Table 2 and Table 3). Interestingly, an additional attending surgeon was found to be associated with 0.48 days longer stay following TJA (5.82± 2.17 vs. 6.3 ± 3.1 (*p* = 0.028)) (Table 1) and 0.51 days longer stay following TKA (5.95 ± 2.2 vs. 6.46 ± 2.58 (*p* = 0.049)) (Table 3).

When comparing outcomes with a varying number of surgeons scrubbed in (Table 4, Table 5 and Table 6), surgeries performed by fewer surgeons were related with a shorter surgical duration and an increased need for blood units following TJAs and THAs (Table 4 and Table 5). This was not true of TKAs (Table 6), where the number of surgeons was not found to affect any of the measured parameters.

In THA, an increase in either the number of attending surgeons or in the number of surgeons in total increased the chance of the surgery to be performed through the direct anterior approach (Table 2 and Table 5). When this approach was chosen, surgery was longer (1:45 ± 00:26, 1:45 ± 00:20 and 1:17 ± 0:27 for anterior, lateral and posterior approaches, respectively, *p* < 0.001). However, when looking into the effect of the number of surgeons on surgical length for each approach separately, the number of scrubbed-in surgeons did not influence the surgical length, implying that the anterior approach by itself was responsible for the longer surgery. The approach chosen did not influence the rate of intraoperative complications related to surgical technique (intraoperative fracture or suboptimal implant alignment), 5.6% for direct anterior, 7.5% for lateral and 1% for posterior approach respectably (*p* = 0.3). A further analysis including only anterior approach cases was carried out to detect any difference in outcomes between the single attending and two attendings groups. This analysis revealed no difference in surgical time (1 h 45 min ± 27 min vs. 1 h 47 min ± 24 min, *p* = 0.65), length of stay (4.27 ± 1.6 vs. 5.67 ± 4.52 *p* = 0.1), blood units given (senior 0.07 ± 0.25 units vs. 0.03 ± 0.26 units, *p* = 0.57) and intraoperative complications (3.33% vs. 1.7% *p* = 1). This was also true for the lateral approach when analyzed separately.

The most frequents intraoperative complication for both hips and knees was a periprosthetic fracture (three femoral and two acetabular fractures during THAs and two femoral and one tibial plateau fracture during TKAs). Other intraoperative complications included malposition of implants (in one femoral component and one acetabular component in THAs and in one knee implant), one vascular injury (TKA) necessitating bypass and graft by a vascular surgeon and one case of transient hypotension during femoral preparation in THA.

There were five hip revisions and two knee revisions overall. Four revisions following TJAs were due to a periprosthetic joint infection (PJI) (three revision THAs and one TKR revision). Two patients presented with PJI 4- and 8-weeks following THA, respectively, and were treated with irrigation and debridement due to patients’ co-morbidities (the second patient had a metastatic renal cell carcinoma and died during hospitalization from additional complications). The third THA patient who presented with deep infection 13 weeks following surgery underwent staged revision. One knee revision was due to superficial wound infection 6 weeks following surgery. No implant exchange was required. Infection rate correlated significantly with the overall number of scrubbed surgeons, with rates of 3.9%, 0.6% and 0% for four, three and two scrubbed-in surgeons, respectively (*p* = 0.009). Other revisions were performed for late peri prosthetic fractures (two THAs) and for stiffness (one TKA).

## 4. Discussion

As the demand for TJAs continues to be on the rise [1], there is an increasing pressure on health systems to expand the quantity of surgeries performed. This may lead to a conflict of interests, as more surgeons need to be trained while, at the same time, the training itself may potentially lengthen surgeries and increase complication rates. Adding to the complexity is the initiation of national joint registries, urging hospitals to reduce surgical time and complications and increase patient satisfaction rates [2]. In this current single-center study, we sought to evaluate whether an additional scrubbed attending surgeon assisting in TJAs was superior to a resident, and if increasing the overall number of surgeons led to reduced operating time and complications.

No evidence of a shorter operative time was found when comparing surgeries performed by one or two attending surgeons for both THAs and TKAs. Our results contradict the outcomes reported by Woolson et al. [12] and by Robinson et al. [4], who described longer surgical times for surgeries in which residents performed a major part of the surgeries. Of note, in our study, the second (and above) scrubbed-in co-surgeon, whether an attending or a resident, was assisting only, and not the primary surgeon. Haughom et al. [13] also reported a longer surgical duration when residents were involved; however, their data was derived from a national registry and the level of involvement cannot be standardized. Furthermore, our findings suggest that the addition of scrubbed surgeons did not reduce the surgical time in TKAs and was even found to be a risk factor for a lengthier surgery in THAs. This finding can perhaps be explained by the possible need for more surgeons in complex surgeries.

The direct anterior approach (DAA) for THA has recently had a more widespread uptake throughout the world. Proponents of the DAA claim that patients’ rehabilitation and return to daily activities is hastened, and satisfaction is reported to be high [14,15,16,17,18]. Several studies, however, have demonstrated a longer surgical duration and a higher risk of complications when compared with other approaches, and have suggested that this is the result of a learning curve for the technique [19,20,21]. During the study period, the anterior approach was adopted and utilized selectively by all four TJA surgeons at our institution. Our results demonstrate that in THAs performed via the DAA, more surgeons were scrubbed in, whether an attending or trainees, and it was indeed associated with a lengthier mean surgical duration, probably due to the suggested learning curve. We cannot rule out that more surgeons were scrubbed in as part of their learning of the DAA. Nevertheless, the rate of intraoperative complications was comparable between approaches.

Intraoperative complications in TJAs may lead to increased morbidity and reduced patient satisfaction [22]. Interestingly, neither us nor other researchers who examined the effects of residents’ involvement in TJAs found it to be a risk factor for surgical complications [2,4,12,13,23]. This can be explained by the careful supervision by the leading surgeon throughout the procedure. In a recently published work, Nahhas et al. [24] reported that patients are not fully comfortable with trainee involvement in their TJA surgeries. Our data can perhaps aid in patients’ education as it demonstrates no additional risk in trainees’ involvement.

Arthroplasty surgeries, although considered to be highly beneficial, are associated with substantial costs to healthcare systems globally, and the number of TJAs is projected to continue to show substantial growth [25]. The fact that the presence of an attending co-surgeon was not found to improve the outcomes, either perioperatively or in the following year, may suggest that this model does not offer any advantages in terms of cost-effectiveness.

Operating room traffic was found to be a risk factor for periprosthetic joint infections, because the bacterial counts of airborne microorganisms increasewith increased activity levels within the OR [26,27]. While Panahi et al. [28] found that the circulating nurse and surgical implant representatives constituted the majority of OR traffic, it is likely that an increased number of surgeons in the room also increases the number of airborne microorganisms. Indeed, we found a significantly higher rate of postoperative infections when more surgeons were scrubbed in, with a rise in infection rates when over three surgeons took part in the surgery. However, this finding should be interpreted with caution, as the total rate of infections was merely 0.67%.

We acknowledge the several limitations of this study, inherent to the retrospective cohort design and data collection. Firstly, there was no standardization of the part that each surgeon fulfilled during surgery. Secondly, staff allocation for surgeries was largely random, depending on the available personnel, and not based on the level of training. Thirdly, the lack of physician assistants and fellows in our ORs may also influence the results. Fourthly, we have not looked at patient-reported functional outcomes and satisfaction, which might have shown a possible benefit to either group. Lastly, we did not monitor or control the competence level of the trainees who participated in each surgery, and this may potentially affect the surgical time. Nevertheless, the strength of this study is its relatively large cohort of patients treated in a single teaching hospital, by a small number of fellowship-trained arthroplasty surgeons.

## 5. Conclusions

The involvement of several attending surgeons in TJAs was not found to reduce surgical time, while resident involvement was not related with worse patient outcomes. These findings can potentially aid in TJA cost reduction, while at the same time promote resident training without compromising patient’s safety.

## Figures and Tables

**Table 1 jpm-12-01683-t001:** Total joint arthroplasties by number of attending surgeons.

		Total (n = 597)	One Attending Surgeon (n = 339)	Two Attending Surgeons (n = 258)	*p* Value
Age (years), average (SD)		70.54 (9.27)	71.42 (8.6)	69.48 (9.98)	0.008
Male gender, n (%)		225 (37.7)	127 (37.5)	98 (38)	0.932
Body mass index, average (SD)		30.11 (5.43)	30.01 (5.43)	30.22 (5.42)	0.66
Current smoking, n (%)		64 (10.7)	33 (9.7)	31 (12)	0.233
ASA score, n (%)	1	18 (3)	6 (1.8)	12 (4.7)	0.089
	2	367 (61.5)	205 (60.5)	162 (62.8)	
	3	212 (35.5)	128 (37.8)	84 (32.6)	
Diabetes mellitus, n (%)		156 (26.1)	91 (26.8)	65 (25.2)	0.707
Malignancy, n (%)		75 (12.6)	52 (15.3)	23 (8.9)	0.024
Rheumatoid arthritis, n (%)		13 (2.2)	10 (2.9)	3 (1.2)	0.165
General anesthesia, n (%)		498 (83.4)	282 (83.2)	216 (83.7)	0.286
Length of surgery (hours), average (SD)		1:39 (0:29)	1:41 (0:29)	1:37 (0:28)	0.152
Residents aiding in surgery, average (SD)		1.31 (0.538)	1.47 (0.54)	1.11 (0.47)	*p* < 0.001
Blood units, average (SD)		0.09 (0.435)	0.12 (0.5)	0.06 (0.34)	0.119
Length of stay (days), average (SD)		6.04 (2.62)	5.82 (2.17)	6.3 (3.1)	0.028
Intraoperative complications, n (%)		15 (2.5)	9 (2.7)	6 (2.3)	1
Revision surgery within a year, n (%)		7 (1.2)	4 (1.2)	3 (1.2)	1

**Table 2 jpm-12-01683-t002:** Total hip arthroplasties by number of attending surgeons.

		Total (n = 239)	One Attending Surgeon (n = 117)	Two Attending Surgeons (n = 122)	*p* Value
Age (years), average (SD)		68.72 (10.47)	70.09 (9.6)	67.39 (11.11)	0.046
Male gender, n (%)		108 (45.2)	50 (42.7)	58 (47.5)	0.52
Body mass index, n (%)		28.88 (5.32)	28.78 (5.26)	28.97 (5.39)	0.8
Current smoking, n (%)		15.5% (37)	12.8% (15)	22 (18)	0.22
ASA score, n (%)	1	11 (4.6)	2 (1.7)	9 (7.4)	0.12
	2	148 (66.5)	72 (61.5)	76 (62.3)	
	3	80 (33.5)	43 (36.8)	37 (30.3)	
Diabetes mellitus, n (%)		42 (17.6)	22 (18.8)	20 (16.4)	0.73
Malignancy, n (%)		37 (15.5)	20.5% (24)	13 (10.7)	0.048
Rheumatoid arthritis, n (%)		4 (1.7)	3 (2.6)	1 (0.8)	0.36
General anesthesia, n (%)		224 (93.7)	111 (94.9)	113 (92.6)	0.36
Length of surgery (hours), average (SD)		1:32 (00:29)	1:32 (0:28)	1:33 (0:29)	0.7
Cemented stem, n (%)		79 (33.1)	47 (40.2)	32 (26.2)	0.028
Approach, n (%)	Anterior	89 (37.2)	27 (23.1)	62 (50.8)	*p* < 0.001
	Lateral	40 (16.7)	36 (30.8)	4 (3.3)	
	Posterior	107 (44.8)	51 (43.6)	56 (45.9)	
	Anterolateral	3 (1.3)	3 (2.6)	3 (1.3)	
Residents aiding in surgery, average (SD)		1.32 (0.54)	1.5 (0.52)	1.14 (0.5)	*p* < 0.001
Blood units, average (SD)		0.1 (0.48)	0.15 (0.6)	0.05 (0.31)	0.088
Length of stay (days), average (SD)		5.88 (2.96)	5.61 (2.11)	6.14 (3.58)	0.165
Intraoperative complications, n (%)		9 (3.8)	4 (3.4)	5 (4.1)	1
Revision surgery within a year, n (%)		5 (2.1)	3 (2.6)	2 (1.6)	0.678

**Table 3 jpm-12-01683-t003:** Total knee arthroplasty by number of attending surgeons.

		Total (n = 358)	One Attending Surgeon (n = 222)	Two Attending Surgeons (n = 136)	*p* Value
Age (years), average (SD)		71.75 (8.16)	72.12 (7.95)	71.16 (8.49)	0.28
Male gender, n (%)		117 (32.7)	77 (34.7)	40 (29.4)	0.353
Body mass index, n (%)		30.95 (5.34)	30.68 (5.42)	31.36 (5.23)	0.26
Current smoking, n (%)		7.5% (27)	18 (8.1)	9 (6.6)	0.608
ASA score, n (%)	1	7 (2)	4 (1.8)	3 (2.2)	0.464
	2	219 (61.2)	133 (59.9)	86 (63.2)	
	3	132 (36.9)	85 (38.3)	47 (34.6)	
Diabetes mellitus, n (%)		114 (31.8)	69 (31.1)	45 (33.1)	0.726
Malignancy, n (%)		38 (10.6)	28 (12.6)	10 (7.4)	0.157
Rheumatoid arthritis, n (%)		9 (2.5)	7 (3.2)	2 (1.5)	0.492
General anesthesia, n (%)		274 (76.5)	171 (77)	103 (75.7)	0.699
Length of surgery (hours), average (SD)		1:44 (0:28)	1:46 (0:29)	1:41 (0:26)	0.15
Residents aiding in surgery, average (SD)		1.31 (0.54)	1.45 (0.55)	1.09 (0.43)	*p* < 0.001
Blood units, average (SD)		0.09 (0.407)	0.1 (0.44)	0.07 (0.36)	0.56
Length of stay (days), average (SD)		6.14 (2.36)	5.95 (2.2)	6.46 (2.58)	0.049
Revision surgery within a year, n (%)		2 (0.6)	1 (0.5)	1 (0.7)	1
Intraoperative complications, n (%)		6 (1.7)	5 (2.3)	1 (0.7)	0.414

**Table 4 jpm-12-01683-t004:** Demographics, surgical characteristics and outcomes stratified by overall number of surgeons on total joint arthroplasties.

		Two Surgeons (n = 202)	Three Surgeons (n = 344)	Four Surgeons (n = 51)	*p* Value
Age (years), average (SD)		71.61 (8.85)	70.18 (9.22)	68.73 (10.77)	0.075
Male gender, n (%)		78 (38.6)	128 (37.2)	19 (37.3)	0.946
Body mass index, average (SD)		29.65 (5.51)	30.34 (5.24)	30.21 (6.36)	0.395
Current smoking, n (%)		22 (10.9)	39 (11.3)	3 (5.9)	0.202
ASA score, n (%)	1	6 (2)	10 (2.9)	2 (3.9)	0.037
	2	107 (53)	228 (66.3)	32 (62.7)	
	3	89 (44)	106 (30.8)	17 (33.3)	
Diabetes mellitus, n (%)		52 (25.7)	90 (26.2)	14 (27.5)	0.969
Malignancy, n (%)		32 (15.8)	11 (38)	5 (9.8)	0.218
Rheumatoid arthritis, n (%)		5 (2.5)	7 (2)	1 (2)	0.938
General anesthesia, n (%)		169 (83.7)	285 (82.8)	44 (86.3)	0.528
Length of surgery (hours), average (SD)		1:37 (0:34)	1:40 (0:27)	1:38 (0:26)	0.036
Blood units, average (SD)		0.16 (0.58)	0.06 (0.33)	0.08 (0.4)	0.033
Length of stay (days), average (SD)		5.8 (2.2)	6.12 (2.32)	6.45 (5)	0.194
Revision surgery within a year, n (%)		1 (0.5)	3 (0.9)	3 (5.9)	0.004
Intraoperative complications, n (%)		5 (2.5)	7 (2)	3 (5.9)	0.261

**Table 5 jpm-12-01683-t005:** The effect of the number of surgeons on total hip arthroplasties.

		Two Surgeons (n = 67)	Three Surgeons (n = 146)	Four-Five Surgeons (n = 26)	*p* Value
Age (years), average (SD)		70.87 (9.87)	67.58 (10.35)	69.54 (11.99)	0.095
Male gender, n (%)		31 (46.3)	68 (46.6)	9 (34.6)	0.517
Body mass index, average (SD)		28.12 (5.87)	29.06 (4.96)	29.65 (5.96)	0.399
Current smoking, n (%)		12 (17.9)	23 (15.8)	2 (7.7)	0.306
ASA score, n (%)	1	20 (3)	7 (4.8)	2 (7.7)	0.117
	2	34 (50.7)	97 (66.4)	17 (65.4)	
	3	31 (46.3)	42 (28.8)	7 (26.9)	
Diabetes mellitus, n (%)		12 (17.9)	24 (16.4)	6 (23.1)	0.712
Malignancy, n (%)		17 (25.4)	16 (11)	4 (15.4)	0.026
Rheumatoid arthritis, n (%)		1 (1.5)	2 (1.4)	1 (3.8)	0.657
General anesthesia, n (%)		62 (92.5)	139 (95.2)	23 (88.5)	0.354
Length of surgery (hours), average (SD)		1:26 (0:29)	1:34 (0:28)	1:43 (0:29)	0.023
Cemented stem, n(%)		35 (52.2)	36 (24.7)	8 (30.8)	*p* < 0.001
Approach, n (%)	Anterior	11 (16.4)	63 (43.2)	15 (57.7)	*p* < 0.001
	Lateral	14 (20.9)	25 (17.1)	1 (3.8)	
	Posterior	39 (58.2)	58 (39.7)	10 (38.5)	
	Anterolateral	3 (4.5)	0	0	
Blood units, average (SD)		0.27 (0.77)	0.03 (0.23)	0.08 (0.39)	0.002
Length of stay (days), average (SD)		5.79 (2.13)	5.74 (2.22)	6.89 (6.46)	0.185
Revision surgery within a year, n (%)		1 (1.5)	2 (1.4)	2 (7.7)	0.107
Intraoperative complications, n (%)		2 (3)	4 (2.7)	3 (11.5)	0.088

**Table 6 jpm-12-01683-t006:** The effect of the number of surgeons on total knee arthroplasties.

		Two Surgeons (n = 135)	Three Surgeons (n = 198)	Four Surgeons (n = 25)	*p* Value
Age (years), average (SD)		71.98 (8.32)	72.09 (7.79)	67.88 (9.51)	0.048
Male gender, n (%)		47 (34.8)	60 (30.3)	10 (40)	0.497
Body mass index, average (SD)		30.41 (5.18)	31.31 (5.24)	30.8 (6.83)	0.364
Current smoking, n (%)		10 (7.4)	16 (8.1)	1 (4)	0.751
ASA score (%)	1	4 (3)	1.5% (3)	0	0.211
	2	73 (54.1)	131 (66.2)	15 (60)	
	3	58 (43)	64 (32.3)	10 (40)	
Diabetes mellitus, n (%)		40 (29)	66 (33.3)	8 (32)	0.776
Malignancy, n (%)		15 (11.1)	22 (11.1)	1 (4)	0.538
Rheumatoid arthritis, n (%)		4 (3)	5 (2.5)	0	0.685
General anesthesia, n (%)		107 (79.3)	146 (73.7)	21 (84)	0.598
Length of surgery (hours), average (SD)		1:42 (0:34)	1:44 (0:24)	1:53 (0:21)	0.188
Blood units, average (SD)		0.1 (0.45)	0.08 (0.38)	0.08 (0.4)	0.875
Length of stay (days), average (SD)		5.8 (2.24)	6.39 (2.36)	6 (2.86)	0.075
Revision surgery within a year, n (%)		0	1 (0.5)	1 (4)	0.047
Intraoperative complications, n (%)		3 (2.2)	3 (1.5)	0	0.704

## Data Availability

Not applicable.

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
