# Peer review of "Single Surgeon versus Co-Surgeons in Primary Total Joint Arthroplasty: Does “Two Is Better than One” Apply to Surgeons?"

_jpm, 2022, doi:10.3390/jpm12101683_

Round 1

Reviewer 1 Report

The presented article is quite interesting and well done. 

The topic is interesting for many orthopedic surgeons. 

The study-group is large enough and the results are well done using statistical methods. 

The conclusion is done using the main literature. 

Author Response

September 26, 2022

Re: Manuscript ID: jpm-1924780

Article title: “Single-Surgeon versus Co-surgeons in Primary Total Joint Arthroplasty: Does “Two is Better Than One” Apply to Surgeons”

Dear Editor,

Please see below for a point-by point response to the reviewers’ comments and concerns. In addition, please find attached an amended version of the manuscript. All changes are marked in green highlight.

We are very grateful to the reviewers for their positive and helpful suggestions, and we feel that the quality of the manuscript has been significantly improved as a result.

We look forward to seeing our manuscript in your journal.

Response to the reviewer #1 comments:

Comment 1.  Were all four surgeons equally distributed in both groups?

Answer:  A further statistical analysis was carried out to examine the four senior surgeons and their distribution between the two study groups. The analysis showed an equal distribution among the two groups (p=0.776). This was also true separately for total knee arthroplasty (p=0.42) and total hip arthroplasty (p=0.57). The following sentence was added to the results section: “The four senior surgeons were equally distributed between the two groups (p=0.776 for the entire cohort, p=0.42 for TKA, p=0.57 for THA).”  

Comment 2.  Were all approaches, in particular anterior and lateral, equally distributed in both groups?

Answer: Looking into the different approaches, there was a notable difference in the distribution of cases performed via both the anterior and the lateral approaches, between the two study groups (Table 2).  We therefore performed two additional analyses. The first analysis included only anterior approach cases to look for any significant differences between the two groups (single attending and two attendings). This analysis revealed no differences in: surgical time (1h 45 min ± 27 min vs. 1h 47 min ±24 min, p=0.65), length of stay (4.27±1.6 vs 5.67 ±4.52 p=0.1), blood units given (senior 0.07 ± 0.25 units vs 0.03 ± 0.26 units, p=0.57) and intraoperative complications (3.33% vs 1.7% p= 1). The second analysis was performed including only lateral approach cases, looking for any differences between the two groups. Again, we found no significant differences in : surgical time (1h 45 min ± 19 min vs. 1h 40 min ± 21 min, p=0.66), length of stay (7.5±3.93 vs 6.5 ±1.9 p=0.75), blood units given (0.17 ± 0.59 units vs 0.0 units, p=0.56) and intraoperative complications (7.5% vs 25% p= 0.32). The following paragraph was therefore added to the results section where relevant: “A further analysis including only anterior approach cases was carried out to detect any difference in outcomes between the single attending and two attendings groups. This analysis revealed no difference in surgical time (1h 45 min ± 27 min vs. 1h 47 min ±24 min, p=0.65), length of stay (4.27±1.6 vs 5.67 ±4.52 p=0.1), blood units given (senior 0.07 ± 0.25 units vs 0.03 ± 0.26 units, p=0.57) and intraoperative complications (3.33% vs 1.7% p= 1). This was also true for the lateral approach when analyzed separately.”

Comment 3.  Were the three knee systems equally distributed?

Answer:  As specified in the manuscript, three systems were used. A further analysis was performed and showed that the three systems were equally distributed between the groups (single attending vs. two attendings or more), p=0.178 (Pearson Chi-square).The following sentence was added to the results section: In the TKA cohort, the three different knee systems used were equally distributed between the single attending and two attendings groups (p=0.178). We would also like to emphasize that all knee replacements were performed using a similar surgical technique (mechanical alignment through a standard medial parapatellar approach) which also reduces the potential bias derived from the surgical technique. This is now clarified in the methods section: “In all cases, mechanically aligned TKA was performed through a standard medial parapatellar approach, and cement was used.”

Reviewer 2 Report

Dear authors,

thank you very much for giving me the opportunity to review this work. There is no goldstandard in arthroplasty surgery for an ideal number of surgeons and it seems obvious that a skilled second surgeon could possibly be beneficial. Therefore, this well-structured work is of high clinical interest.

However, the significance of your results might be limited since four different surgeons used four different approaches (hip) and three different systems(knee).

Please extend the data and statistics by giving answers to the following questions.

Were all four surgeons equally distributed in both groups?

-        Were all approaches, in particular anterior and lateral, equally distributed in both groups?

-        Were the three knee systems equally distributed?

Best wishes

Author Response

September 26, 2022

Re: Manuscript ID: jpm-1924780

Article title: “Single-Surgeon versus Co-surgeons in Primary Total Joint Arthroplasty: Does “Two is Better Than One” Apply to Surgeons”

Dear Editor,

Please see below for a point-by point response to the reviewers’ comments and concerns. In addition, please find attached an amended version of the manuscript. All changes are marked in green highlight (The revised manuscript is attached).

We are very grateful to the reviewers for their positive and helpful suggestions, and we feel that the quality of the manuscript has been significantly improved as a result.

We look forward to seeing our manuscript in your journal.

Response to the reviewer #1 comments:

Comment 1.  Were all four surgeons equally distributed in both groups?

Answer:  A further statistical analysis was carried out to examine the four senior surgeons and their distribution between the two study groups. The analysis showed an equal distribution among the two groups (p=0.776). This was also true separately for total knee arthroplasty (p=0.42) and total hip arthroplasty (p=0.57). The following sentence was added to the results section: “The four senior surgeons were equally distributed between the two groups (p=0.776 for the entire cohort, p=0.42 for TKA, p=0.57 for THA).”  

Comment 2.  Were all approaches, in particular anterior and lateral, equally distributed in both groups?

Answer: Looking into the different approaches, there was a notable difference in the distribution of cases performed via both the anterior and the lateral approaches, between the two study groups (Table 2).  We therefore performed two additional analyses. The first analysis included only anterior approach cases to look for any significant differences between the two groups (single attending and two attendings). This analysis revealed no differences in: surgical time (1h 45 min ± 27 min vs. 1h 47 min ±24 min, p=0.65), length of stay (4.27±1.6 vs 5.67 ±4.52 p=0.1), blood units given (senior 0.07 ± 0.25 units vs 0.03 ± 0.26 units, p=0.57) and intraoperative complications (3.33% vs 1.7% p= 1). The second analysis was performed including only lateral approach cases, looking for any differences between the two groups. Again, we found no significant differences in : surgical time (1h 45 min ± 19 min vs. 1h 40 min ± 21 min, p=0.66), length of stay (7.5±3.93 vs 6.5 ±1.9 p=0.75), blood units given (0.17 ± 0.59 units vs 0.0 units, p=0.56) and intraoperative complications (7.5% vs 25% p= 0.32). The following paragraph was therefore added to the results section where relevant: “A further analysis including only anterior approach cases was carried out to detect any difference in outcomes between the single attending and two attendings groups. This analysis revealed no difference in surgical time (1h 45 min ± 27 min vs. 1h 47 min ±24 min, p=0.65), length of stay (4.27±1.6 vs 5.67 ±4.52 p=0.1), blood units given (senior 0.07 ± 0.25 units vs 0.03 ± 0.26 units, p=0.57) and intraoperative complications (3.33% vs 1.7% p= 1). This was also true for the lateral approach when analyzed separately.”

Comment 3.  Were the three knee systems equally distributed?

Answer:  As specified in the manuscript, three systems were used. A further analysis was performed and showed that the three systems were equally distributed between the groups (single attending vs. two attendings or more), p=0.178 (Pearson Chi-square).The following sentence was added to the results section: In the TKA cohort, the three different knee systems used were equally distributed between the single attending and two attendings groups (p=0.178). We would also like to emphasize that all knee replacements were performed using a similar surgical technique (mechanical alignment through a standard medial parapatellar approach) which also reduces the potential bias derived from the surgical technique. This is now clarified in the methods section: “In all cases, mechanically aligned TKA was performed through a standard medial parapatellar approach, and cement was used.”

Round 2

Reviewer 2 Report

Dear authors,

thank You very much for your additional analysis.

Your work truly shows that an additional will not not a significant negative effect (neither a positive) which is important for any teaching hospital.

As you already pointed out a second prospective randomized trial would be great.

Best regards